# IRFL: Image Recognition of Figurative Language

**Ron Yosef, Yonatan Bitton, Dafna Shahaf**
The Hebrew University of Jerusalem
{ron.yosef, yonatan.bitton,dshahaf}@cs.huji.ac.il

## Abstract

Figures of speech such as metaphors, similes, and idioms are integral parts of human communication. They are ubiquitous in many forms of discourse, allowing people to convey complex, abstract ideas and evoke emotion. As figurative forms are often conveyed through multiple modalities (e.g., both text and images), understanding multimodal figurative language is an important AI challenge, weaving together profound vision, language, commonsense and cultural knowledge.

In this work, we develop the Image Recognition of Figurative Language (IRFL) dataset. We leverage human annotation and an automatic pipeline we created to generate a multimodal dataset, and introduce two novel tasks as a benchmark for multimodal figurative language understanding. We experimented with state-of-the-art vision and language models and found that the best (22%) performed substantially worse than humans (97%). We release our dataset, benchmark, and code[1], in hopes of driving the development of models that can better understand figurative language.

## 1 Introduction

Figures of speech such as metaphors, similes, and idioms are integral parts of human communication. They are ubiquitous in many forms of discourse, allowing people to convey complex, abstract ideas, compare situations, provke thought and evoke emotion (Lakoff and Johnson, 1980; Hoffman and Kemper, 1987; Roberts and Kreuz, 1994; Fussell and Moss, 1998). Figurative language research often focuses on text alone; however, figurative language is often conveyed through *multiple* modalities (usually text and images) – for example, in areas such as social media, advertising, and news.

Figure 1 shows two social media posts that require multimodal figurative understanding. In the

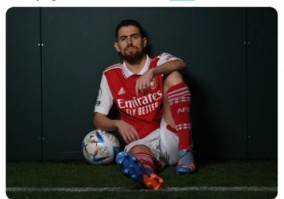 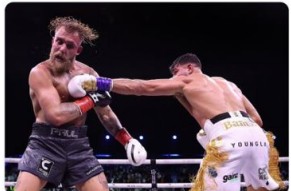

Figure 1: Two social media posts that require multimodal figurative understanding to comprehend. The left photo depicts a soccer player who left his struggling club. The right photo shows an amateur boxer who lost a boxing match to a professional boxer.

left image, the caption reads "Jumped off the sinking ship just in time", and the image shows a soccer player who has just left his struggling club. The right image, captioned "A performing clown", shows a famous YouTuber losing a boxing match to a professional boxer.

Due to its integral part in human communication, detecting and understanding multimodal figurative language could play an important role in various multimodal challenges, such as hate-speech detection in memes (Das et al., 2020), fact checking (Yao et al., 2022), sentiment analysis (Soleymani et al., 2017), humor recognition (Reyes et al., 2012; Shahaf et al., 2015; Schifanella et al., 2016), and tasks focusing on the mental state of social media users (Yadav et al., 2020; Cheng and Chen, 2022).

Vision and Language Pre-Trained Models' (VL-PTMs) understanding of figurative language combined with images has not been thoroughly explored, partly due to the lack of large-scale datasets. In this work, we introduce the **IRFL dataset** (Image Recognition of Figurative Language) of idioms, metaphors, and similes with matching images – both figurative and literal. We developed a pipeline to collect candidate figurative and literal images and annotated them via crowdsourcing.

Next, we used the dataset to design two novel

---

[1] https://irfl-dataset.github.io/

tasks, **multimodal figurative language detection** and **multimodal figurative language retrieval**, to assess the figurative-language capabilities of VL-PTMs. The detection task is to choose the image that best visualizes the figurative phrase. See Figure 2 for an example for an idiom, a metaphor, and a simile, with the correct answers highlighted.

As we noticed that VL-PTMs tend to select images containing *objects* that appear in the figurative phrase, we designed a second task targeting this behavior. In the multimodal figurative language retrieval task, the goal is to rank figurative images higher than images with objects from the phrase.

We experiment with several VL-PTMs and find that the best model (22% accuracy) fails to reach human performance (97%). We also find that generative models have difficulties generating figurative images for idiomatic phrases.

We hope our dataset and benchmarks will drive the development of multimodal models that can better understand figurative language, closing the gap with human performance. More broadly, metaphorical reasoning is strongly tied to problem-solving and creativity; we believe such models, able to see analogies between situations that share very little on the surface, could greatly advance the field.

## 2 Background

We start with a short introduction to the main types of figurative language (Lakoff and Johnson, 1980; Paul, 1970; Philip, 2011).

A **metaphor** is a comparison between concepts that makes us think of the target concept in terms of the source concept. For example, in "You're a peach!", a person is equated with a peach, suggesting that they are pleasing or delightful.

A **simile** also compares two things, often introduced by "like" or "as". A simile is open when the shared properties are not explicitly revealed ("Her heart is like a stone"), and closed when they are ("Her heart is hard as stone").

An **idiom** is a group of words with a non-literal meaning that can not be understood by looking at its individual words. E.g., "We're on the same page" means "Agreeing about something".

Understanding figurative language requires commonsense, general knowledge, and the ability to map between domains. Understanding idioms, in particular, requires profound language and cultural knowledge (Paul, 1970; Philip, 2011).

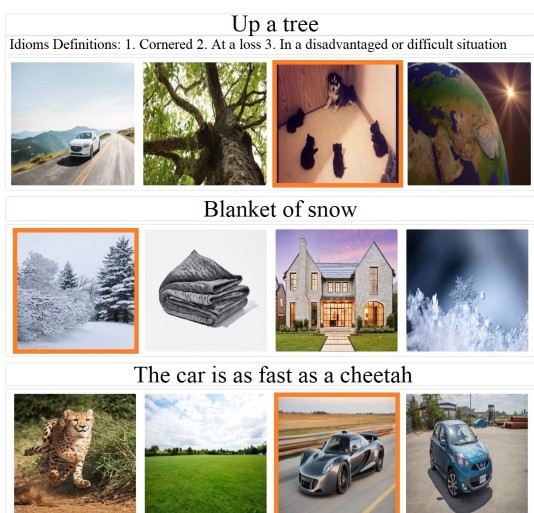

Figure 2: Examples of the multimodal figurative language detection task for idiom, metaphor, and simile. The input is a figurative phrase and four candidate images (for idiom, we also show the definition). The correct answer is marked with an orange square.

## 3 The IRFL Dataset

Our goal is to create a dataset with idioms, metaphors, and similes paired with figurative and literal images. This dataset can then serve as a benchmark to evaluate Vision and Language models on multimodal figurative language.

**Labels.** Initially, we intended to have our annotators label images "literal" or "figurative". However, after initial experimentation with the data generated by our pipeline, we realized the necessity of a more nuanced classification system. Hence, we introduced two additional categories.

The first new category, "Figurative+Literal," encompasses images that express the figurative meaning of an expression while also maintaining some aspects of the literal interpretation. The second, "Partial Literal," includes images that visualize some (literal) elements or objects from the expression.

Table 1 illustrates our categories for the expression "Touch wood". For example, an image of someone literally touching wood while crossing his fingers for luck is classified as Figurative+Literal. This distinction also allows us to later perform a richer analysis of model performance.

### 3.1 Pipeline: Idioms

We collected 628 idioms from the MAGPIE corpus (Haagsma et al., 2020) of idiomatic expressions. The MAGPIE corpus contains 56,622

| Idiom: Touch wood | | | | |
|---|---|---|---|---|
| **Definitions:** 1) Hopefully 2) Said while touching something wooden, to avert superstitious bad luck from what has just been said | | | | |

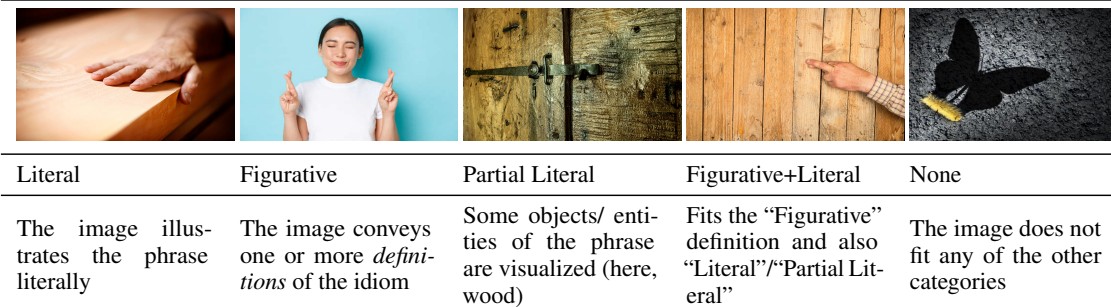

| Literal | Figurative | Partial Literal | Figurative+Literal | None |
|---|---|---|---|---|
| The image illustrates the phrase literally | The image conveys one or more *definitions* of the idiom | Some objects/ entities of the phrase are visualized (here, wood) | Fits the "Figurative" definition and also "Literal"/"Partial Literal" | The image does not fit any of the other categories |

Table 1: The table shows the different categories of the relation between an image and a phrase, along with matching images for the idiom "Touch wood". Workers were guided to choose the most suitable relation category by a scheme tree that illustrates the correct thinking process (Figure 5, Appendix A.1).

crowdsourced potentially idiomatic expressions, covering $1,756$ unique idioms that appear in at least two of the following dictionaries: Wiktionary, Oxford Dictionary of English Idioms, and UsingEnglish.com. After collecting the idioms, we feed them into our pipeline.

Our pipeline consists of four main steps, illustrated in Figure 3. Given an idiom, we first get its definitions from online dictionaries and parse them into search queries (§3.1.1). Second, we search for candidate images using the queries. Third, we filter the images and select the best $k$ literal and figurative candidates for annotation (§3.1.3). Lastly, we annotate the different images via crowdworkers.

### 3.1.1 Searching for Images

Our goal is to find literal and figurative images for each idiom from the MAGPIE dataset. Searching for an idiom using image search often results in

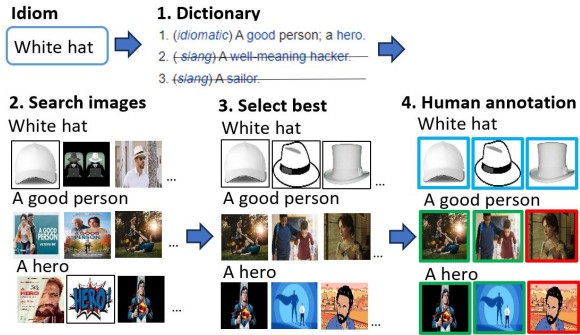

Figure 3: The flow of our idiom pipeline: getting definitions, looking for image candidates using the idiom and its definitions, filtering an selecting candidate images. In the human annotation stage, blue represents Literal, Green – Figurative, and red – None.

literal images. To find figurative images, we need to understand the *meaning* of the idiom; however, the MAGPIE dataset does not contain idiom definitions, so we crawl them from online dictionaries (Wiktionary definitions tagged with 'figurative' or "idiomatic"[2]; if no such definitions exist, we try the Oxford Dictionary).

For example, in Figure 3, the idiom "white hat" nd is defined as "A good person; a hero" (tagged with "idiomatic"), and also as "a sailor" and "A well-meaning hacker" (tagged with "slang").

We split concatenated definitions (e.g., "A good person; a hero" is split into two definitions). In some rare cases, a definition may be another idiom, and then we replace that idiom with its definitions.

We then searched Google images for the idioms and their definitions, taking up to 20 images per search query. Images were searched with "Safe-Search" flag "on", and in "United States" region.

### 3.1.2 Image Filtering

We noted that many of the retrieved images contained the search query in textual form. We used optical character recognition (OCR) tool EasyOCR to extract text from the images, and TextBlob to correct spelling errors the OCR made. We then filtered images that contained objects or entities from the idiom or its definitions in textual form (50% of the images). Such images are problematic because they may cause the model to select an image solely based on its textual signal. Following this filter, 15% of the resulting images contained mostly

---

[2]We also construct search queries from untagged definitions. Even though untagged definitions are rare (<3%), they are typically idiomatic.

|  | Fig. | Fig. Lit. | Lit. | Part. Lit. | None | |
|---|---|---|---|---|---|---|
| # | 1970 | 751 | 434 | 487 | 2638 | 6697 |
| 3-maj | 100% | 100% | 100% | 100% | 100% | 94% |
| 4-maj | 75.5% | 63% | 68% | 63% | 80% | 70% |
| 5-maj | 45% | 33% | 35% | 38% | 53% | 43% |
| Mean | 3.1 | 1.2 | 0.7 | 0.8 | 4 | - |
| Median | 2 | 0 | 0 | 0 | 4 | - |

Table 2: IRFL statistics on 628 idioms. The majority of the images are related to the figurative phrase, most images are Figurative. (k-maj means k-majority)

text. To tackle this problem, we used OCR (See Appendix A.2) to remove images with more than a couple of words, as well as images with more than 30% of their space containing text.

For the remaining images, we calculated the matching score of each image with its phrase and search query using ViLT. Top-$k$ images with a high "phrase-image" score (that passed a threshold, see Appendix A.3) were tagged as potentially literal. We chose the top $k$ images with the highest "definition-image" score as Figurative candidates.

### 3.1.3 Human Annotation

We hired Amazon Mechanical Turk (AMT) workers to annotate the relation between each idiom and its candidate images using the user interface seen in Appendix A.1 (Figure 6). Five workers annotated each image in batches of five images per sample. They received a payment of $0.15 per sample, which resulted in an average hourly wage of $15. We created a qualification test[3] to select quality annotators and provided them with an interactive training platform[4] to understand the task and the different categories better.

We split the annotation process into batches with an average size of 60 idioms per batch. After each batch, we provided each worker with a personal profile page (Appendix A.1, Figure 7) to view their statistics and some examples where their choice was different from the majority of workers.

Full annotation results and statistics are presented in Table 2. Despite the subjective nature of the task, in 94% of the instances, there was a majority of 3 workers or more out of 5 compared to a random chance of 29%.

[3]https://irfl-dataset.github.io/mturk/image/qualification
[4]https://irfl-dataset.github.io/mturk/image/train

### 3.2 Pipeline: Metaphors and Similes

We collected 35 textual metaphors and 142 textual similes, compiled from online lists. Generating search queries from definitions (to find figurative images) is a central part of our pipeline for idioms (Section 3.1). However, idioms are fixed expressions, but metaphors and similes are much more flexible, as the number of possible comparisons between two things is vast.

For this reason, we had to adapt our pipeline. For metaphors, we asked three expert annotators to agree upon definitions. For similes, we use the simile itself and the target concept with the shared property ("fast") as search queries to find figurative images. For literal images that serve as distractors, we use the source and target without the shared property. In some cases, the target concept images are inadequate literal distractors (an image of a car might still be considered figurative for the simile "The car is as fast as a cheetah"). To solve this problem, we include the *antonym* of the property ("A slow car").

**Annotation.** As the number of images was relatively small, we had two experts from our team manually annotate images. We obtained 1107 figurative and 1816 partial literal images for similes, 333 figurative and 729 partial literal for metaphors (the other categories were less relevant for the specific data generated by our pipeline).

## 4 Experiments

### 4.1 Multimodal Figurative Language Detection Task

The **Multimodal Figurative Language Detection Task** evaluates VL-PTMs' ability to choose the image that best visualizes the meaning of a figurative expression. Figure 2 shows an example of the task for an idiom, a metaphor, and a simile.

Our goal was to create a difficult and diverse task representing the richness of our dataset (Section 3). We constructed 810 "mixed" task instances for idioms, metaphors, and similes. Each "mixed" instance contains four candidates: one is the correct answer, partially literal distractors, and random images.

Idiom instances have 1-2 partially literal distractors. Simile instances contain two literal distractors, one of the target concept without the compared property or with its antonym visualized, and one of the source concept. Metaphor "mixed" instances

consist of between 1-3 partially literal distractors.

**Zero-Shot Baselines.** We evaluate several state-of-the-art vision-and-language models. We use four versions of CLIP models (Radford et al., 2021): RN50, ViT-B/32, ViT-L/14, and RN50x64/14 with 100M, 150M, 430M, and 620M parameters, respectively. We use the official implementations of ViLT (Kim et al., 2021), BLIP (Li et al., 2022), CoCa ViT-L-14 (Yu et al., 2022), and BLIP2 (Li et al., 2023b). We evaluate all models with their default hyper-parameters, except ViLT on idioms, due to its maximum sequence length of 40 tokens.

The models encode the figurative phrase and the image, producing a matching score for each pair. We choose the image with the highest score as the one best matches the expression.

We also experimented with multimodal chatbot models, including LLaVA (Liu et al., 2023), InstructBLIP (Dai et al., 2023), OpenFlamingo (Awadalla et al., 2023), and Otter (Li et al., 2023a). We found that the first two do not support our setting, as they can not handle questions about multiple images; the latter two do support the setting, but did not seem to understand the task, returning mostly nonsense answers.

**Supervised Models.** We train a supervised model for figurative classification of idioms. We add a binary classifier on top of pre-trained embeddings to classify whether a given image is figurative. We use CLIP (VIT-B/32) model, concatenating the textual idiom embedding to the visual image embedding, followed by a classifier that produces a matching score. A score above 0.5 is labeled "Figurative". We use the Adam optimizer (Kingma and Ba, 2014) with a learning rate of 0.001, batch size of 12, and train for 7 epochs. We run the fine-tuned model on the multimodal figurative language detection (§4.1) task using the model's matching score. We train the binary classifier on 4790 images, making sure the training data does not contain any of the images or idioms that appear in the task. We repeat five experiments with different random seeds for each task and take the mean score and std.

### 4.1.1 Human Evaluation

We asked annotators that did not work on previous IRFL tasks to solve the multimodal figurative language detection task. Each instance of the "mixed" multimodal detection task was annotated by 5 annotators, and the correct answer was chosen by the majority. We find that human performance on

| Categories | Idiom | | Metaphor | Simile | |
|---|---|---|---|---|---|
| | Fig. | Fig. Lit. | | Cl. | Op. |
| Humans | 97 | 90 | 99.7 | 100 | |
| CLIP-VIT-L/14 | 17 | 56 | 25 | 52 | 40 |
| CLIP-VIT-B/32 | 16 | 44 | 23 | 45 | 38 |
| CLIP-RN50 | 14 | 37 | 27 | 47 | 35 |
| CLIP-RN50x64 | **22** | 56 | **30** | 52 | 41 |
| BLIP | 18 | **57** | 22 | **66** | **44** |
| BLIP2 | 19 | 53 | 19 | 57 | 40 |
| CoCa ViT-L-14 | 17 | 53 | 18 | 45 | 33 |
| ViLT | - | - | 23 | 40 | 33 |
| # Phrases | 48 | 30 | 35 | 142 | 137 |
| # Tasks | 135 | 65 | 333 | 277 | 277 |

Table 3: Zero-shot models performance on the IRFL "mixed" multimodal figurative language detection task. There are two columns for idioms and similes. "Closed" and "Open" refers to the simile type. "Figurative" and "Figurative+Literal" refer to the correct image category. Numbers are the percentage of instances annotated correctly. Models fail to reach human performance across all figures of speech.

the data sampled for all figures of speech ranges between $90\% - 100\%$ (Table 3). Additionally, in $82\% - 99\%$ of the instances, there was an agreement between at least four annotators compared to a random chance of $6\%$. Samples from the validation process are presented in Appendix A.5.

### 4.1.2 Results and Model Analysis

Zero-shot results on the "mixed" multimodal figurative language detection task are presented in Table 3. The best model achieved $22\%$, $30\%$, and $66\%$ accuracy on the idioms[5], metaphors, and similes tasks compared to a random chance of $25\%$. These results suggest that **models do not understand the connection between a figurative phrase and an image as humans do.** We next conduct a fine-grained analysis to examine if models failed because they do not see any connection to the figurative images or rather because they prioritize literal connections over figurative ones.

**Models prefer partially literal images over figurative ones.** We analyze the models' choices on the "mixed" multimodal figurative language detection task and found that in all models, a partially literal distractor was selected in $92\% - 100\%$ of the instances where the models failed across all figures of speech (idioms, metaphors, and similes). This shows that models prefer partially literal images

---

[5]Idioms were passed along with their definitions as input.

| Categories | Fig. | | | | | | Fig. Lit. | |
|---|---|---|---|---|---|---|---|---|
| Candidates | 2 | | 4 | | 6 | | 4 | |
| Random | 50 | | 25 | | 16.6 | | 25 | |
| CLIP-VIT-L/14 | 64 | **87** | **46** | **71** | **33** | 53 | 76 | 86 |
| CLIP-VIT-B/32 | 61 | 84 | 38 | 67 | 30 | 53 | 65 | 82 |
| CLIP-RN50 | 56 | 75 | 30 | 60 | 24 | 46 | **78** | 86 |
| CLIP-RN50x64 | **67** | 79 | 38 | 67 | 27 | 51 | 69 | 85 |
| BLIP | 57 | 79 | 30 | 62 | 19 | 51 | 72 | 88 |
| BLIP2 | 58 | 75 | 25 | 58 | 14 | 40 | 75 | 82 |
| COCA ViT-L-14 | 62 | 82 | 39 | **71** | 32 | **60** | 68 | **91** |

Table 4: Zero-shot models performance on different configurations of the multimodal figurative language detection task, idioms with random candidates. Numbers are % instances annotated correctly. The left column of each pair shows the score for the idiom alone as input, and the right column shows the score for the idiom and definitions. Models fail to reach human performance.

| Categories | Metaphors | | Similes | | | |
|---|---|---|---|---|---|---|
| | | | Closed | | Open | |
| Candidates | 2 | 4 | 2 | 4 | 2 | 4 |
| CLIP-VIT-L/14 | 87 | 72 | **99** | 97 | 97 | **96** |
| CLIP-VIT-B/32 | 86 | 73 | **99** | 97 | 97 | 95 |
| CLIP-RN50 | 83 | 66 | **99** | 97 | 98 | 94 |
| CLIP-RN50x64 | **88** | **76** | 98 | 96 | 96 | 94 |
| BLIP | 76 | 58 | **99** | **98** | 98 | 94 |
| BLIP2 | 72 | 55 | **99** | 93 | 95 | 88 |
| CoCa ViT-L-14 | 83 | 71 | **99** | 97 | **99** | **96** |
| ViLT | 72 | 53 | 96 | 91 | 97 | 89 |

Table 5: Zero-shot models performance on the multimodal figurative language detection task, metaphors and similes with random candidates. Numbers are % instances annotated correctly. Models' performance on similes is competitive with humans.

over figurative ones. We find the case of idioms to be particularly interesting. Models receive a relatively long prompt (idiom+definitions), and often choose an image that is a literal interpretation of only 1-2 words from the prompt.

**Models partially understand the figurative connection between idioms and images.** To examine whether models can comprehend a figurative connection between an image and an idiom, we experiment with random candidates and several configurations of the multimodal figurative language detection task (Table 4). When provided with an idiom and its definitions as input, the accuracy on the Figurative category ranges between $75\% - 87\%$ with 2 candidates and $58\% - 71\%$ with 4 candidates. These results are above chance level but still below human performance on the "mixed" task.

When given the idiom alone as input, the accuracy ranges between $56\% - 67\%$ with 2 candidates and $25\% - 46\%$ with 4 candidates. These results suggest that models partially understand the figurative connection between idioms and images. We see a significant performance drop with all models when the number of candidates increases.

In the Figurative+Literal category, with only the idiom as input, models registered an accuracy of $65\% - 78\%$ with 4 candidates. This performance significantly exceeds the accuracy recorded on the Figurative category with 2 and 4 candidates. The performance increase can be explained by the fact that Figurative+Literal images have both a literal and figurative connection to the phrase.

**Models understand metaphors but fail to reach human performance.** Table 5 shows the models' performance on metaphors with random candidates. The accuracy of all models on the Figurative category with 2 candidates is $72\% - 88\%$, and $53\% - 76\%$ with 4 candidates. We see a significant performance drop with all models when the number of candidates increases. The results suggest that models can understand metaphors but fail to reach human performance.

**Models understand similes well.** Table 5 shows the models' performance on the similes with random candidates. The accuracy of all models on the Figurative category with 2 candidates is $95\% - 99\%$, and $88\% - 98\%$ with 4 candidates. Models' performance is competitive with that of humans, and the models maintain their performance when increasing the number of candidates. In contrast to the multimodal figurative language detection task with random images, the "mixed" task shows a performance gap between closed and open similes due to open similes concealing the compared property, making it harder for the model to choose the figurative image. Analyzing the "mixed" task results on closed similes, we found that figurative images scored higher than source concept images in $52\% - 74\%$ of cases across all models.

Additionally, source concept images scored higher than target concept distractor images in $51\% - 70\%$ of cases. This pattern suggests a model prioritization for simile images: firstly, target concept images with the compared property, then source concept images, and finally, target con-

| Categories | Fig. | Fig. Lit. |
|---|---|---|
| Zero-Shot Idiom | 5% | 36% |
| Supervised Idiom | 46.2% ± 3.6 | 41.1% ± 3 |
| Zero-Shot Idiom + Def. | 16% | 41% |
| Supervised Idiom + Def | 58% ± 4.2 | 49% ± 2.6 |

Table 6: The performance of Supervised and Zero-shot models, both when provided with only idioms and when provided with idioms along with their definitions. During training, the supervised model received the same input configuration as it was tested on. Compared to zero-shot results, the supervised results are about $3.6 - 9\times$ better in the figurative category, while figurative-literal results improved by $13 - 20\%$.

cept images lacking the compared property.

**Fine-tuning improves figurative understanding and reduces literal preference.** The supervised model results are presented in Table 6. Previously we did not display the models' performance on the "mixed" task when taking the idiom alone as input due to their poor performance ($5\% - 7\%$ accuracy). However, when training on idioms alone, the supervised model scored a mean accuracy of $46.2\%$, $9\times$ the zero-shot score of $5\%$. This large performance increase might suggest that VL-PTMs representation of an idiom encodes its definitions.

Training and testing with the idiom and its definitions as input resulted in a mean accuracy of $58\%$ compared to $16\%$ in the Zero-shot configuration. After analyzing the supervised model results, we found that its literal preference has improved significantly. In $41\% \pm 4.3$ of the instances where the model failed, a partially literal distractor was selected compared to $96\%$ in the zero-shot configuration. Along with this improvement in literal preference, Figurative+Literal category accuracy raised from $41\%$ in zero-shot to $49\%$. These results show that models can improve their preference for partially literal images and recognize idiomatic figurative connections better via training. Moreover, the results suggest that the data is a useful training signal for our task.

We have discovered that VL-PTMs tend to prefer partially literal images. In the next section, we design a task to tackle this issue.

## 4.2 Multimodal Figurative Language Retrieval Task

The **Multimodal Figurative Retrieval Task** examines VL-PTMs' preference for figurative images. Given a set of figurative and partially literal

### ruffle someone's feathers

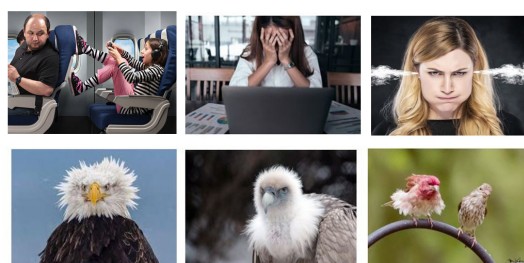

Figure 4: Example of multimodal figurative language retrieval task for the idiom "ruffle someone's feathers" (to unease, cause discomfort to someone). The task is to rank the figurative images above the partial literal ones, based on the images' matching score with the idiom.

images, the task is to rank the images using the model-matching score such that the figurative images are ranked higher, and calculate the precision at $k$, where $k$ is the number of figurative images in the input.

Figure 4 shows an example of the task for the idiom "ruffle someone's feathers". We wish to have images of people causing discomfort ranked higher than pictures of birds and feathers. This task provides an opportunity for a deeper probe into how the model comprehends figurative language in terms of its preferences.

In this task, we use the same baselines and training methods mentioned in the previous task. We train the supervised model on 3802 images, making sure the training data does not contain any of the images or idioms that appear in the task.

### 4.2.1 Results and Model Analysis

Zero-shot results are presented in Table 7. We evaluate all figurative phrases that have both Figurative and Partial Literal images. Models' scores on the preference task are low ($<61\%$). We expect models with proper figurative preference to achieve better results. Models' success in the Figurative+Literal category can be attributed to the literal connections of the Figurative+Literal images.

The supervised model achieved a score of $68 \pm 3.8$ in the Figurative category, almost double the zero-shot score of CLIP-ViT-B/32 (36). Additionally, the score in the Figurative+Literal category was improved by $10 \pm 2.25$ points. These results align well with the observation that the multimodal figurative language detection task supervised model, which was trained using the same method on a different training set, also showed

| Categories | Idioms | | Metaphors | Similes | |
|---|---|---|---|---|---|
| | Fig. Lit. | Fig. | | Cl. | Op. |
| CLIP-VIT-L/14 | 57 | 37 | 26 | 44 | **34** |
| CLIP-VIT-B/32 | 54 | 36 | 22 | 38 | 30 |
| CLIP-RN50 | 54 | 37 | 25 | 38 | 31 |
| CLIP-RN50x64 | **61** | **39** | **29** | 43 | 32 |
| BLIP | 58 | **39** | 24 | **54** | 33 |
| BLIP2 | 57 | **39** | 22 | 42 | 29 |
| CoCa ViT-L-14 | 56 | 36 | 20 | 39 | 24 |
| ViLT | - | - | 25 | 34 | 28 |
| # Phrases | 94 | 149 | 35 | 142 | 137 |

Table 7: Models performance on the multimodal figurative language retrieval task, the scoring metric is mean precision at $k$, where $k$ is the number of figurative images. There are two columns for idioms and similes. "Closed" and "Open" refers to the simile type. "Figurative" and "Figurative+Literal" refer to the correct image category. The results are low, we expect better results from models with proper figurative preferences.

substantially moderate literal preference. Table 8 shows the fine-tuned model results.

| Categories | Fig. | Fig. Lit. |
|---|---|---|
| Zero-Shot | 36 | 54 |
| Supervised | $68 \pm 3.8$ | $64 \pm 2.25$ |

Table 8: Supervised models performance. The scoring metric is mean precision at $k$, where $k$ is the number of figurative images. of five experiments. Compared to zero-shot results, the supervised results increased by about 88% and 16% in the figurative and figurative+literal categories.

### 4.3 Generative Models Analysis

In our work so far, we focused on finding existing images matching a figurative expression. We now explore the question of whether generative models can *generate* figurative images. We sampled 15 idioms from the IRFL dataset and experimented with the idioms and their definitions as input to Dall-E (Ramesh et al., 2021) and Stable Diffusion (Rombach et al., 2022). We annotated 345 generated images and found that generative models failed to generate figurative images for given idioms, generating literal images instead. When provided with the definitions as input, the models had some more success in creating figurative images. Statistics on the generated images can be seen in Table 9. We also included the percentage of images from each category found by our pipeline.

| Categories | Dall-E | | Stable Diffusion | | IRFL | |
|---|---|---|---|---|---|---|
| Figurative | 0 | 42.5 | 0 | 11 | 4 | 46 |
| Figurative+Literal | 0 | 10 | 5 | 1 | 20 | 6 |
| Literal | 31 | 0 | 17 | 0 | 35 | 0 |
| Partial Literal | 48 | 2 | 42 | 2.5 | 23 | 1.5 |
| None | 19 | 44 | 27 | 85 | 4 | 43 |
| Number | 48 | 120 | 59 | 118 | 69 | 126 |

Table 9: The table is double-columned, the first column describes the percentage of images generated by idioms, and the second column describes the percentage of images generated by the idioms' definitions. The results show that our pipeline extracted more Figurative, Figurative+Literal, and Literal images and fewer None images than the generative models.

The results show that our pipeline extracted more Figurative, Figurative+Literal, and Literal images and fewer None images than the generative models managed to generate.

## 5 Related Work

**Idioms.** Several papers have examined pre-trained LMs' ability to represent idioms. Shwartz and Dagan (2019) found that LMs' representation of idiomatic expressions was of lower quality than that of literal ones. Chakrabarty et al. (2022) introduced a narrative understanding benchmark focused on interpreting figurative language and found that pre-trained LMs struggle to perform well in zero-shot and few-shot settings. To the best of our knowledge, Vision and Language Pre-trained models (VL-PTMs) understanding of idioms has not been investigated until this work.

**Metaphors and Similes.** Recently there have been several works exploring the ability of VL-PTMs to understand similes and metaphors, and several datasets have been introduced (Zhang et al., 2021; hen Liu et al., 2022; Chakrabarty et al., 2023; Hwang and Shwartz, 2023). These datasets often focus on different types of images (memes, politics, advertising), sometimes containing synthetic images (Akula et al., 2022). In contrast, we use natural images from a search engine. In addition, our tasks introduce the new aspect of retrieval.

**Commonsense.** Commonsense is a topic of increasing interest. Particularly relevant lines of work deal with abstractions, associations, and analogies (Mitchell, 2021; Ji et al., 2022; Bitton et al., 2022), all required for understanding figurative language. For example, understanding "as stubborn as a mule"

requires the commonsense (false) association between mules and stubbornness.

# 6 Conclusions and Future Work

In this work we introduced IRFL, a dataset of Figurative and Literal images for idioms, metaphors, and similes. We developed two novel tasks as a benchmark for multimodal figurative language understanding. Our experiments demonstrate that the tasks are easy for humans and challenging for state-of-the-art vision and language models. We publish our dataset, benchmark, and code.

In the future, we hope to extend this work to other modalities and different forms of figurative speech. In addition, there are interesting cross-cultural connections between figurative expressions. For example, the English expression "cost an arm and a leg" (meaning expensive) has a corresponding expression in French: "Coûter les yeux de la tête" (literally, cost the eyes of the head). Adapting our ideas to languages other than English, taking advantage of such connections, is another promising direction.

We believe that multimodal figurative language is an essential aspect of human communication that is under-explored in AI; we hope that this work will encourage the development of multimodal models that can better understand figurative language.

More broadly, metaphorical reasoning is strongly tied to problem-solving and creativity; we believe that models that can see analogies between situations that share very little on the surface could find many potential applications.

# 7 Limitations

Our dataset focuses on English idioms. As translation of figurative expressions is a particularly delicate task, it is not straightforward to expand our dataset to other languages, and further research is needed to explore the effectiveness of our pipeline to other languages. In addition, our method heavily relies on sources of figurative expressions, their definitions, and the image search engine.

## Acknowledgements

We want to thank Nir Sweed for his valuable feedback. We would also like to thank Nissim Barzilay for his contribution to the collection of figurative and literal images for metaphors and similes. This work was supported by the European Research Council (ERC) under the European Union's Horizon 2020 research and innovation program (grant no. 852686, SIAM, Shahaf).

In memory of the more than one thousand victims of the horrific massacre carried out by Hamas terrorists on October 7th, 2023.

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

# A  Appendix

## A.1  Annotation UI

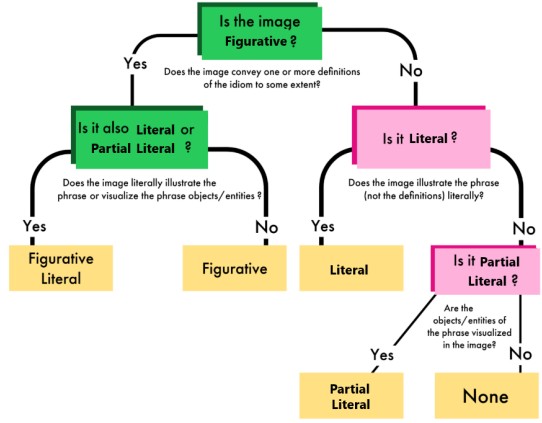

Figure 5: The scheme tree that was provided to annotators to illustrate the correct thinking process.

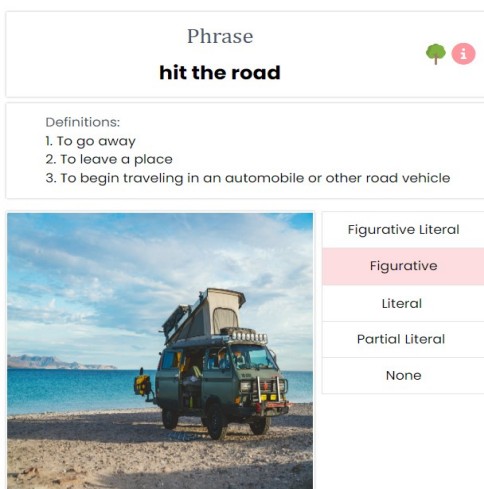

Figure 6: The UI used to annotate the automatic pipeline candidate images. Annotators need to choose the category that best describes the relationship between the idiom and the image.

## A.2  Documents Filter

In an effort to minimize the number of images dominated by text, we filtered out images containing more than a few words, which accounted for 15% of the total. Despite this, certain images like documents, books, and contracts managed to bypass our OCR-based filters, representing 2% of the total images. To address this issue, we developed a filter using the ViLT model (Kim et al., 2021). This filter calculates an image's matching score with the prompts "a document", "a page of a book", or "a contract" and removes it if the total score surpasses a set "document" threshold. To find this threshold, we conducted a grid search on 20 sampled images at each point in the distribution of $-30, -25, -20, -15, -10, -5, 0, 5, 10, 15, 20, 25, 30$ categorizing each as a "document" or "non-document". The $(20, 15)$ range showed the best results, so we conducted a more dense grid search within this range and found the best threshold to be 18.77 with a TPR of 100% and an FPR of 1%.

## A.3  Literal Threshold

We conducted two grid searches on images that passed the OCR filters and had a "phrase-image" score higher than the "search-query" score to find a literal threshold. We sampled 20 images from each point in the distribution of $-10, -8, -6, -4, -2, 0, 2, 4, 6, 8, 10,$ and annotated them as "literal" or "non-literal". This distribution aligns with the normal distribution of the images that stand the two criteria mentioned

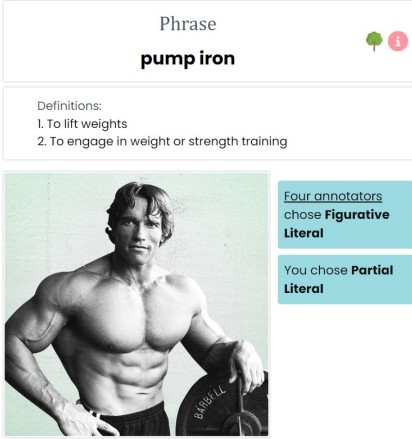

Figure 7: An example of the profile page includes the worker's statistics and some handily picked examples where his choice was distant from a majority of four workers.

**Mturk Leaderboard** 🏆

| Worker | Figurative Literal | Figurative | Literal | Partial Literal | None | Images |
|--------|-------------------|-----------|---------|-----------------|------|--------|
| WORKER_ID | 84.11 | 90.26 | 97.40 | 88.37 | 88.37 | 716 |
| WORKER_ID | 83.48 | 84.07 | 86.59 | 97.92 | 96.35 | 781 |
| WORKER_ID | 97.39 | 87.17 | 84.15 | 77.08 | 95.99 | 781 |
| WORKER_ID | 78.79 | 92.22 | 82.35 | 83.33 | 79.80 | 265 |
| WORKER_ID | 75.00 | 79.29 | 68.25 | 82.50 | 92.03 | 681 |

Figure 8: An example of the profile page includes the worker's statistics and some handily picked examples where his choice was distant from a majority of four workers.

above (Figure 9). We found the $(-2, 2)$ range to result in the best thresholds, so we conducted a more dense grid search in this range. We sampled

30 images from each point in the distribution of $-5, -4, -2, -1, 0, 1, 2, 4, 5,$ and annotated them as "literal" or "non-literal". We chose the threshold of $1.150353$ with a TPR of $86\%$ and FPR of $18\%$.

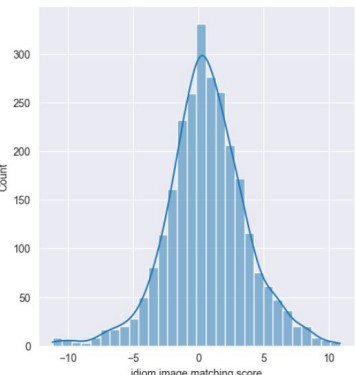

Figure 9: The distribution of the images that passed the OCR filters and had a "phrase-image" score higher than the "search-query" score.

We observed that when the "phrase-image" score is high, we can say that the image is literal with a high probability. However, the reverse is not true, there can be multiple "literal" images with a very low literal score (Figure 10).

### A.4 GenBench Evaluation Card

| Motivation | | | |
|---|---|---|---|
| *Practical* | *Cognitive* | *Intrinsic* ☐ | *Fairness* |
| **Generalisation type** | | | | | |
| *Compo-sitional* ☐ | *Structural* | *Cross Task* | *Cross Language* | *Cross Domain* | *Robust-ness* ☐ |
| **Shift type** | | | |
| *Covariate* | *Label* | *Full* ☐ | *Assumed* |
| **Shift source** | | | |
| *Naturally occuring* | *Partitioned natural* ☐ | *Generated shift* | *Fully generated* |
| **Shift locus** | | | |
| *Train–test* ☐ | *Finetune train–test* ☐ | *Pretrain–train* | *Pretrain–test* |

Table 10: The GenBench evaluation card (Hupkes et al., 2023) for the IRFL Multimodal Figurative Language Detection Task and the Multimodal Figurative Language Retrieval Task.

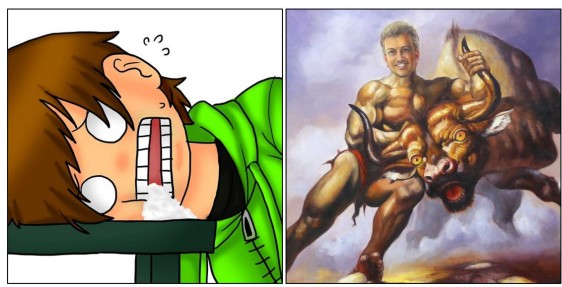

Figure 10: Literal images of the idiom "Foam at the mouth" and the idiom "Take the bull by the horns". Both images have a "phrase-image" score of −9.

## A.5 Understanding Task Samples

### shrinking violet

1. A very shy or timid person, who avoids contact with others if possible

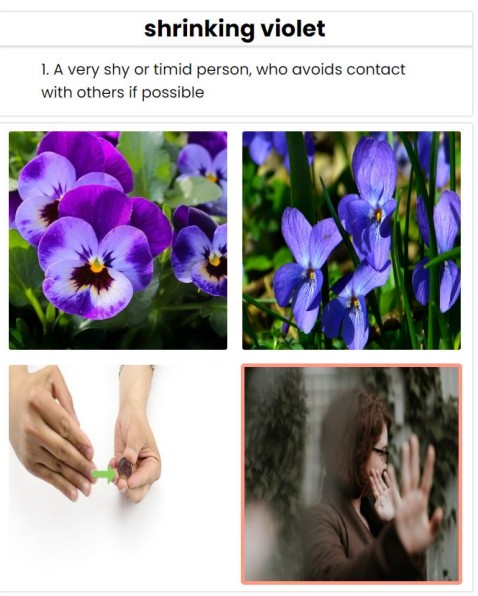

### press the panic button

1. To start to panic

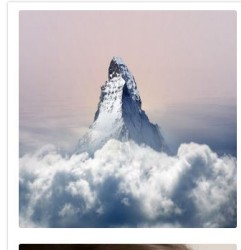 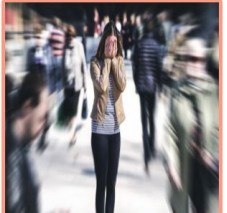
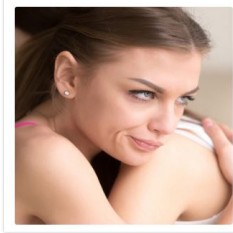 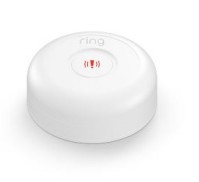

### save someone's skin

1. To save someone's life
2. To prevent an undesirable occurrence

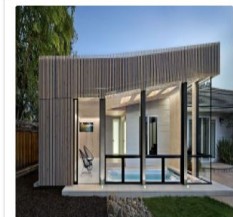 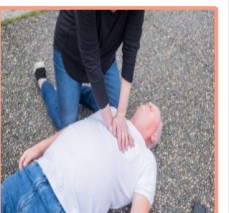
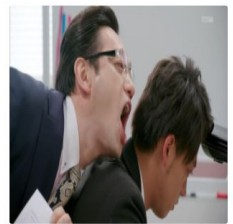 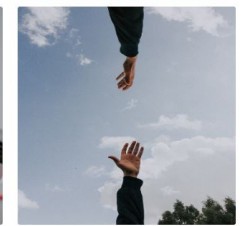

### get the boot

1. To be dismissed from employment
2. To be voted out, evicted, or otherwise made to leave

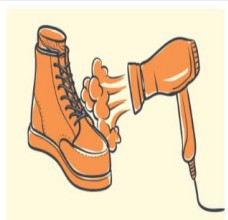 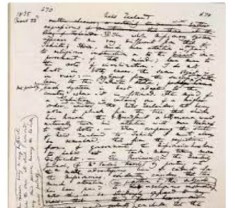
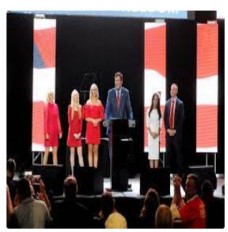 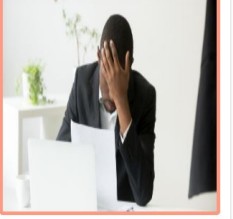

## heart of gold

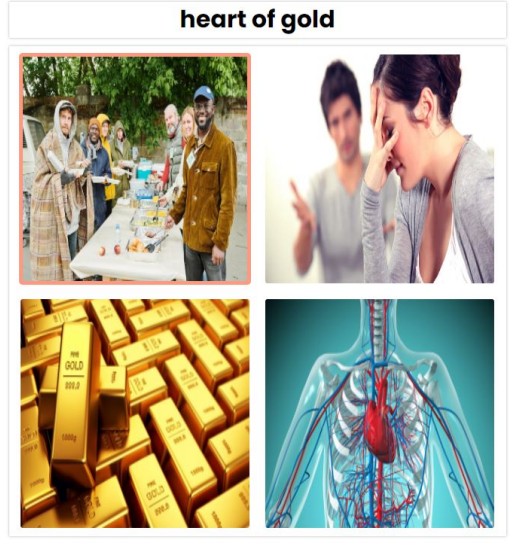

## jungle city

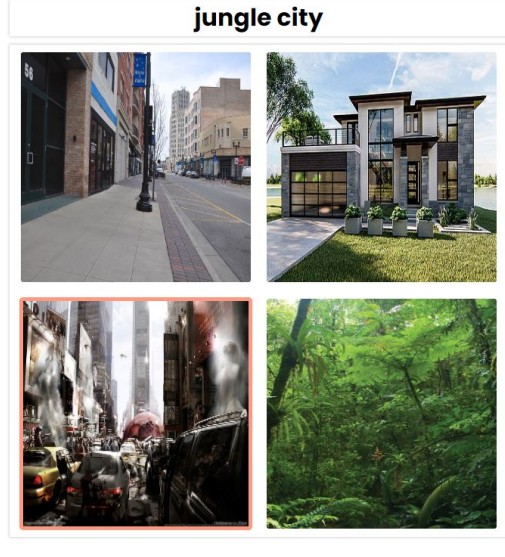

## a night owl

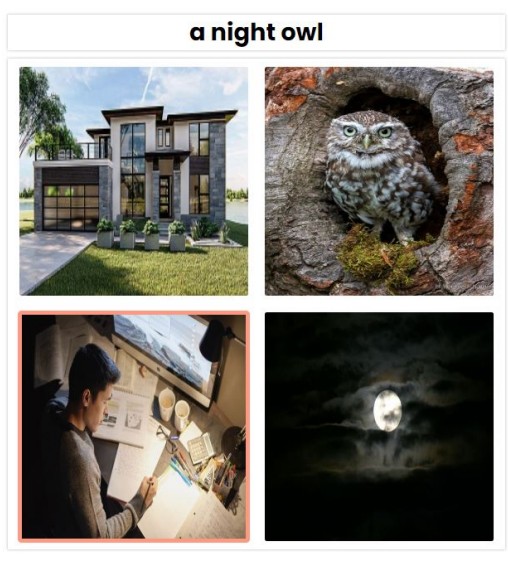

## The juice is as sweet as sugar

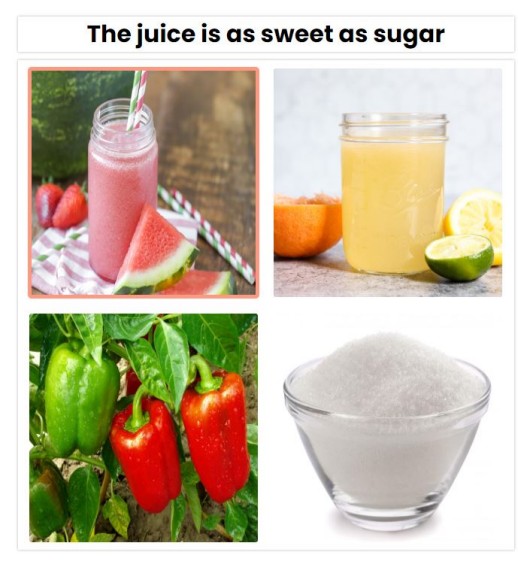

## the car is a rocket

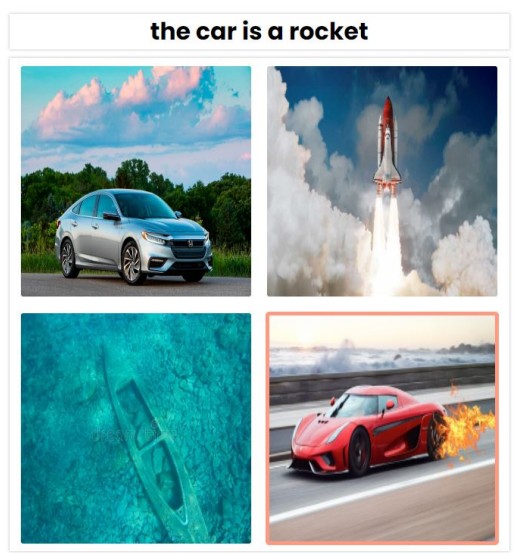

## The dog is as busy as a bee

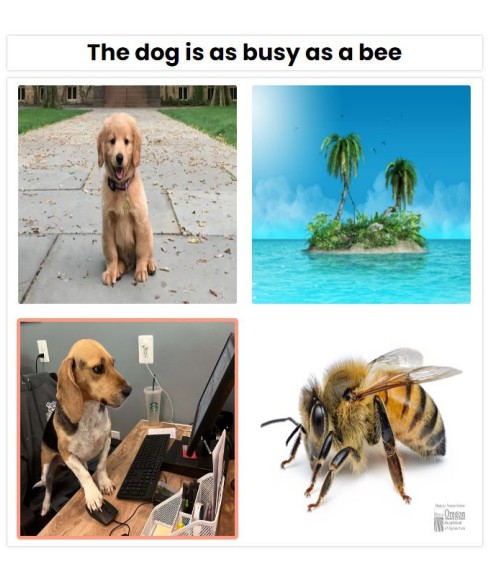

## The frog is as red as a tomato

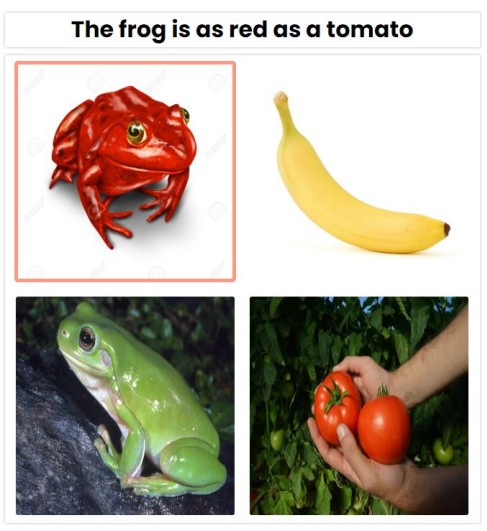

## The milk is as fresh as a daisy

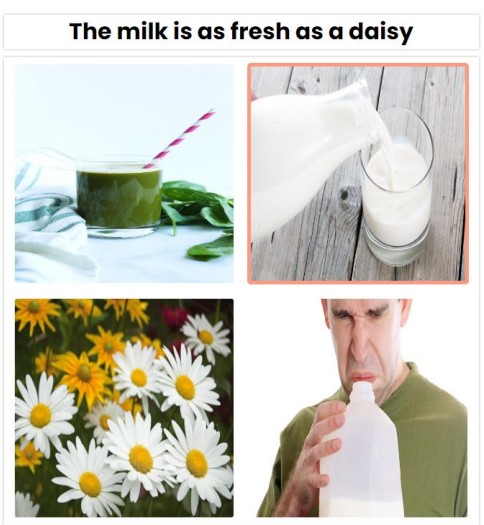