# OpenReview forum: "IRFL: Image Recognition of Figurative Language"
_EMNLP/2023/Conference — EMNLP 2023 Findings_

### Official Review · Reviewer_W39D · 2023-08-05

**Soundness:** 4

**Excitement:**

4: Strong: This paper deepens the understanding of some phenomenon or lowers the barriers to an existing research direction.

**Paper Topic And Main Contributions:**


The paper proposes the "Image Recognition of Figurative Language" dataset to evaluate the understanding of figurative language by vision and language models. The authors leverage human annotation and an automatic pipeline to generate a multimodal dataset and introduce two novel tasks as a benchmark for multimodal figurative understanding.

**Reasons To Accept:**

This article accurately distinguishes the concepts of metaphor, simile, and idiom. Figure 5, the scheme tree, is very useful. In comparison to some of the dataset studies I have seen before, this article starts from the most fundamental elements and avoids sacrificing clear definitions for the sake of quantity (such as conflating metaphors and analogies).


In addition to the existing images matching figurative expressions, the paper explores whether generative models can create figurative images. Sampling 15 idioms from the IRFL dataset, they experiment with Dall-E and Stable Diffusion models. The results indicate that generative models struggle to generate figurative images for given idioms, producing literal images instead. Providing definitions as input improves the performance to some extent, but challenges remain in generating accurate figurative representations.


The paper sheds light on the difficulties in multimodal figurative language understanding, urging further research to develop models that better comprehend figurative language.

**Reasons To Reject:**

The proposed dataset and benchmark tasks may not adequately capture the complexity and diversity of real-world multimodal figurative language understanding scenarios.

**Reproducibility:**

5: Could easily reproduce the results.

**Reviewer Confidence:**

3: Pretty sure, but there's a chance I missed something. Although I have a good feel for this area in general, I did not carefully check the paper's details, e.g., the math, experimental design, or novelty.

---

> ### Author Rebuttal · Authors · 2023-08-29
>
> Thank you for your useful feedback. We are glad you found our formulation clear and accurate.
>
> * “The proposed dataset and benchmark tasks may not adequately capture the complexity and diversity of real-world multimodal figurative language understanding scenarios.”
>
> We completely agree that real-world multimodal figurative language is complex and diverse. While we have not covered the entire landscape, we believe that our dataset covers a significant (and interesting) portion of it: idioms, metaphors, and similes are some of the most common (and well-studied) types of figurative language. For each phrase, we collect multiple images that are related to the phrase in different ways (figurative interpretation, literal interpretation, etc.).
>
> We will add a note to the Future Work section suggesting more types of figurative language to explore, and discuss the challenges they pose (e.g., allusions rely heavily on world knowledge and cultural references), as well as potential additional modalities (such as audio). We hope our work will invoke and catalyze further research in this emerging field.

---

### Official Review · Reviewer_Sk6i · 2023-08-05

**Soundness:** 3

**Excitement:**

2: Mediocre: This paper makes marginal contributions (vs non-contemporaneous work), so I would rather not see it in the conference.

**Paper Topic And Main Contributions:**

This paper presents a new dataset and two tasks to evaluate the figurative recognition capabilities of common vision-language models. The authors also provide an extensive experiment of the dataset with several typical VMLs as well as human annotators. The results show that existing visual language models perform poorly on figurative recognition, which is not surprising.

**Reasons To Accept:**

1. The proposed dataset and benchmark will drive the development of multimodal models that better understand figurative language.
2. A lot of processing and human labeling will give us a high-quality dataset.
3. It is interesting to explore the ability of figurative recognition for visual language models.

**Reasons To Reject:**

1. In my opinion, figurative language understanding should be in a specific context. it is not so valuable to guess the actual meaning of a figurative short phrase. It is more like commonsense reasoning than a vision language understanding task. My suggestion is that you can put the figurative text in a more complex context and then explain the meaning of it, possibly by retrieving related images. Without further context, a figurative expression can have multiple meanings.
2. Some figurative expressions are even very difficult for me to understand. It is because I do not have any relevant knowledge. I can only guess what the original words mean. I think it is the same with vision-language models. As I noted above, commonsense is probably the key issue for your task, so you can test more multi-modal large language models like LLaVA and InstructBLIP.
3. There are no suggestions or solutions for improving the ability to understand figurative language.

**Reproducibility:**

5: Could easily reproduce the results.

**Reviewer Confidence:**

4: Quite sure. I tried to check the important points carefully. It's unlikely, though conceivable, that I missed something that should affect my ratings.

---

> ### Author Rebuttal · Authors · 2023-08-29
>
> Thank you for your constructive comments. We are glad that you appreciate the quality of our dataset and  recognize its potential to drive the development of multimodal models with better figurative understanding.
>
> * “In my opinion, figurative language understanding should be in a specific context. it is not so valuable to guess the actual meaning of a figurative short phrase.”
>
> We agree that understanding figurative language in context is an interesting task, but we argue that the standalone task is valuable and worth exploring in its own right (and indeed, has been studied before).
>
> First, we note that language models have seen figurative language in a myriad different contexts during training; in a sense, we are testing whether they managed to learn the _gist_ of each expression. We also note that figurative language is often used with very little context in real-world scenarios such as social media posts or advertising.
>
> We believe that the fact that our task is easy for humans yet hard for models makes it a good stepping stone towards models that can better understand figurative language, whether with or without context.
>
> * “It is more like commonsense reasoning than a vision language understanding task”
>
> While commonsense is very important for understanding some forms of figurative language (e.g., open similes), the task goes beyond commonsense reasoning. For example, no amount of knowing about jumping and sharks will help one understand what the idiom “jump the shark” means.
>
> In the case of similes and metaphors, the models also need to perform a mapping between the two modalities, understanding the links between them.
>
> * “Some figurative expressions are even very difficult for me to understand. It is because I do not have any relevant knowledge. I can only guess what the original words mean”
>
> As non-native speakers, we can definitely relate. As you noted yourself, we made a significant effort to obtain clear examples of figurative language data through an extensive annotation process. We believe that the high success rate of humans on our benchmark demonstrates the quality of the data.
> (Also, as we noted, language models are likely to have seen those expressions in context, so they have more information than merely relying on guessing the meaning of words)
>
>
> * “you can test more multi-modal large language models like LLaVA and InstructBLIP”
>
> In the paper we tested models that were available a month before the submission; in follow-up work, we are planning to assess the performance of models such as LLaVA, Llama-Adapter, InstructBLIP, etc., as well as release a collab notebook that will allow others to easily test their models on our benchmark.
>
> * “There are no suggestions or solutions for improving the ability to understand figurative language.”
>
> We performed an error analysis, noting that in the vast majority of model failures the partially literal distractor was selected, and that training and fine-tuning significantly improve the results. We will gladly expand on this in the future work section, outlining some of our ideas for improving model performance even further.

---

### Official Review · Reviewer_nddL · 2023-08-12

**Paper Topic And Main Contributions:** 1. Introduce a new question of figura…
**Soundness:** 4

**Excitement:**

4: Strong: This paper deepens the understanding of some phenomenon or lowers the barriers to an existing research direction.

**Questions For The Authors:**

There have been many open-source multimodal LLMs recently that have showcased impressive generalization capabilities. I am curious to know if there are any considerations to assess these models using this multimodal figurative understanding benchmark.

**Reasons To Accept:**

1. The acceptance of this article could potentially catalyze the advancement of multimodal models that possess an enhanced understanding of figurative language. In contrast to existing multimodal datasets that primarily focus on factual question-answering and descriptive contexts, this particular dataset places its emphasis on the diversity, intricacy, and abstraction of expressive forms. Such an endeavor has the potential to stimulate the creation of multimodal models understanding higher levels of abstraction.
2. This work presents clear and robust experiments that illuminate common weaknesses in the current state of VLMs in terms of their comprehension abilities.

**Reasons To Reject:**

There is a mention of the utilization of training data in line 295. However, it is not clear to me where these training data come from. From my understanding, it seems that the data collected in Section 3 is quite limited and primarily constitutes a test-only dataset. It would be better if you could provide further clarification on the training data.

**Reproducibility:**

4: Could mostly reproduce the results, but there may be some variation because of sample variance or minor variations in their interpretation of the protocol or method.

**Reviewer Confidence:**

4: Quite sure. I tried to check the important points carefully. It's unlikely, though conceivable, that I missed something that should affect my ratings.

---

> ### Author Rebuttal · Authors · 2023-08-29
>
> Thank you for your valuable comments. We are glad that you consider our experiments clear and robust, and that you recognize our work’s potential to advance multimodal models’ understanding of higher levels of abstraction.
>
> * “It would be better if you could provide further clarification on the training data”
>
> We are sorry for the confusion. Line 295 refers only to idioms, for which we collected enough data to create both a training set and a test set. For metaphors and similes your understanding is correct. We will clarify this point in the manuscript, including more details about the training data.
>
>
> * “I am curious to know if there are any considerations to assess these models using this multimodal figurative understanding benchmark”
>
> We are aware of the recent multimodal LLMs, and are quite excited about their potential. In the paper we tested models that were available a month before the submission; in follow-up work we are planning to assess the performance of models such as LLaVA, Llama-Adapter, InstructBLIP, etc., as well as release a collab notebook that will allow others to easily test their models on our benchmark.

---

### Meta-Review · Area_Chair_oupx · 2023-09-12

**Recommendation:** 4

**Metareview:**

This paper proposes an innovative new benchmark for assessing the comprehension of figurative language / images, and provides baseline performances of several models. The performance of models tested in the paper (sota at submission time) is far from human performance. It also shows that generative models like Dall-E struggle to generate figurative images.
The paper provides a useful problem description, distinguishing metaphors, similes, and idioms.

---

### Decision · Program_Chairs · 2023-10-07

**Decision:**

Accept-Findings

**Comment:**

This paper proposes an innovative new benchmark for assessing the comprehension of figurative language / images, and provides baseline performances of several models. The performance of models tested in the paper (sota at submission time) is far from human performance. It also shows that generative models like Dall-E struggle to generate figurative images.
The paper provides a useful problem description, distinguishing metaphors, similes, and idioms.